# Association between Beverage Consumption and Sleep Quality in Adolescents

**DOI:** 10.3390/nu16020285

**Published:** 2024-01-18

**Authors:** Lydi-Anne Vézina-Im, Dominique Beaulieu, Stéphane Turcotte, Anne-Frédérique Turcotte, Joannie Delisle-Martel, Valérie Labbé, Lily Lessard, Mariane Gingras

**Affiliations:** 1Département des Sciences de la Santé, Université du Québec à Rimouski (UQAR), Campus de Lévis, 1595 Boulevard Alphonse-Desjardins, Lévis, QC G6V 0A6, Canada; dominique_beaulieu@uqar.ca (D.B.); joannie_delisle-martel@uqar.ca (J.D.-M.); lily_lessard@uqar.ca (L.L.); 2Centre de Recherche du CISSS de Chaudière-Appalaches, 143 Rue Wolfe, Lévis, QC G6V 3Z1, Canada; stephane_turcotte@ssss.gouv.qc.ca (S.T.); anne-frederique.turcotte.cisssca@ssss.gouv.qc.ca (A.-F.T.); 3Collectif de Recherche sur la Santé en Région, 1595 Boulevard Alphonse-Desjardins, Lévis, QC G6V 0A6, Canada; 4Axe Santé des Populations et Pratiques Optimales en Santé, Centre de Recherche du CHU de Québec, 2400 Avenue D’Estimauville, Québec, QC G1E 6W2, Canada; 5CHAU-Hôtel-Dieu de Lévis, 143 Rue Wolfe, Lévis, QC G6V 3Z1, Canada; v_labbe2000@yahoo.ca; 6Chaire Interdisciplinaire sur la Santé et les Services Sociaux pour les Populations Rurales, 1595 Boulevard Alphonse-Desjardins, Lévis, QC G6V 0A6, Canada; 7Direction de Santé Publique, CISSS de Chaudière-Appalaches, 55 Rue du Mont-Marie, Lévis, QC G6V 0B8, Canada; mariane.gingras.cisssca@ssss.gouv.qc.ca

**Keywords:** caffeine, sugar-sweetened beverages, energy drinks, drinking water, sleep, adolescent

## Abstract

The objective of this study was to verify if the consumption of different beverages (such as water, 100% pure fruit juice, and sugar-sweetened beverages (SSBs)) is associated with adolescents’ sleep quality. French-speaking adolescents were recruited in person and online throughout the province of Québec (Canada) from the end of March to early July 2023. Beverage consumption and sleep quality were measured using French versions of validated questionnaires specifically designed for adolescents. A total of 218 adolescents (14–17 years; 55.5% female) completed the online survey. Among caffeinated SSBs, energy drink (r_s_ = −0.16; *p* = 0.0197) and sugar-sweetened coffee (r_s_ = −0.33; *p* < 0.0001) intake was correlated with adolescents’ sleep quality. Energy drink consumption (β = −0.0048; *p* = 0.0005) and being male (β = 0.6033; *p* < 0.0001) were associated with adolescents’ sleep quality. There was an interaction between sugar-sweetened coffee intake and biological sex that was associated with adolescents’ sleep quality (*p* = 0.0053). Sugar-sweetened coffee consumption was correlated with adolescent girls’ abilities to go to bed (r_s_ = −0.21; *p* = 0.0203) and fall asleep (r_s_ = −0.28; *p* = 0.0020), while in boys, it was only significantly correlated with their abilities to go to bed (r_s_ = −0.27; *p* = 0.0069). Public health interventions aimed at adolescent boys should primarily target lowering energy drink consumption, while those aimed at girls should prioritize sugar-sweetened coffee intake to possibly improve their sleep quality.

## 1. Introduction

Recent data suggest that there is an association between diet and sleep, with certain foods favoring good sleep quality and/or a long sleep duration [1,2,3]. Certain beverages might also promote good sleep quality and a long sleep duration. In our previous literature review on water consumption among adolescents, consuming water was associated with a long sleep duration [4]. The intake of 100% pure fruit juice is associated with an earlier weekday bedtime in adolescents [5]. On the opposite hand, sugar-sweetened beverage (SSB) consumption in adolescents is associated with a short sleep duration [6] and social jetlag [7]. Social jetlag occurs when sleep timing is not aligned with a person’s chronotype because of social demands, similar to circadian misalignment [8] (e.g., adolescents having to wake up early on schooldays while they have a late chronotype).

It is likely that water and 100% pure fruit juice consumption replaces SSBs, including caffeinated ones, such as soft drinks like cola, energy drinks, sugar-sweetened coffee and tea (whether cold (e.g., iced tea or coffee) or warm), chocolate milk, and hot chocolate. This could explain why water and 100% pure fruit juice intake seems to be associated with better sleep among adolescents. Adolescents who consume more than one SSB per day tend to drink less than one glass of water per day [9]. Replacing SSBs with water has been found to reduce the risk of obesity in children and adolescents [10,11,12]. Certain interventions are, in fact, aimed at encouraging children and adolescents to substitute SSBs for water [13,14]. It would be interesting to verify if replacing SSBs with water or 100% pure fruit juice could contribute to improve adolescents’ sleep quality.

Caffeine consumption in adolescents is associated with a short sleep duration [15]. A study in New Zealand reported that 59.5% of adolescents mentioned consuming caffeine from warm, soft, and energy drinks after dinner [16]. In Canada, 71.3% of adolescents mentioned having consumed caffeine in the past 12 months [17], and more than 60% of adolescents reported having consumed energy drinks [18]. An overview of systematic reviews reported that 13% to 67% of young people (<18 years) consumed energy drinks in the past year [19]. Energy drink consumption is associated with a short sleep duration among adolescents in Canada [20]. The American Academy of Pediatrics recommends a maximum of 100 mg/day of caffeine in adolescents (12–18 years) [21].

Previous studies have found that sleep quality and beverage consumption in adolescents varies based on biological sex and age. Adolescent girls usually report poorer sleep quality [16] and greater severity of insomnia symptoms [22], while adolescent boys mention consuming more SSBs [23,24], especially energy drinks [16,19,25]. On the other hand, more adolescent girls in New Zealand consumed warm caffeinated beverages, such as coffee, tea, and hot chocolate, while more boys consumed energy drinks after dinner [16]. Older adolescents generally have poorer sleep quality [26] and consume more SSBs [27], including caffeinated ones like sugar-sweetened coffee and tea [28]. Certain differences in sleep between adolescent girls and boys could be explained by age differences [22], as puberty (i.e., a later Tanner stage) is associated with a higher likelihood of experiencing insomnia symptoms in adolescent girls [29]. These results suggest investigating whether biological sex and age can impact the association between beverage consumption and adolescents’ sleep quality.

Most of the research conducted on the association between diet and sleep among adolescents seems to have focused on sleep duration [30], while sleep quality is also important for adolescents’ physical [31] and mental health [32]. The objective of the present study was to fill this gap in the literature by verifying if the consumption of certain beverages (such as water, 100% pure fruit juice, and SSBs, including caffeinated ones) is associated with sleep quality in adolescents. It is hypothesized that water and 100% pure fruit juice intake will be associated with good sleep quality, while the consumption of SSBs, especially caffeinated ones, will be associated with poor sleep quality in adolescents. This information will be useful to develop holistic public health interventions targeting diet and sleep to prevent adolescent obesity [33] and mental health issues [32].

## 2. Materials and Methods

This investigation is part of a larger study on the psychosocial correlates of water consumption among adolescents. Inclusion criteria were (1) being between the ages of 14 and 18 years, (2) speaking French, and (3) living in the province of Québec (Canada). The project was approved by the Research Ethics Committee of the Centre intégré de santé et de services sociaux (CISSS) de Chaudière-Appalaches on 9 February 2023 (2023-1030). Adolescents were recruited in person (e.g., by distributing tracts at sports events) and online (e.g., through advertisements on social media) from the end of March to early July 2023. Twenty CAD 50 gift cards for a local sports store were drawn among participants who completed the survey.

Beverage consumption was measured using the validated French version [34] of the Beverage Questionnaire (BEVQ) [35], a questionnaire developed in the United States to measure the intake of various beverages in children and adolescents. The French version was adapted, such as through the conversion from the imperial to the metric system and the adjustment of some formats to those available in Canada, and validated among a sample of French-speaking adolescents from the province of Québec (Canada) [34]. The BEVQ measures the frequency (never or less than once/week; once a week; 2–3 times/week; 4–6 times/week; once/day; twice/day; 3 times or more/day) and quantity (less than the equivalent of half a can of soda or less than 180 mL or ¾ cup; 250 mL or 1 cup or the equivalent of a small can of soda; equivalent of a regular can of soda or 355 mL or 1½ cup; half a liter or 500 mL or 2 cups; 600 mL or 2½ cups; other, please specify) of beverage consumption in the past month. The SSBs included are regular or non-diet soft drinks; fruit drinks; energy drinks; sports drinks; cold or warm sugar-sweetened tea; cold or warm sugar-sweetened coffee; vitamin water; slushies; flavored milk; hot chocolate; and other SSBs. The BEVQ also measures the intake of 100% pure fruit juice that does not contain any added sugar. Finally, the consumption of the following types of water is measured: non-carbonated and non-flavored water (e.g., tap water); non-carbonated flavored water without added sugar (e.g., herbal tea); carbonated non-flavored water (e.g., plain sparkling water); and carbonated flavored water without added sugar (e.g., non-sweetened lemon-flavored sparkling water).

Sleep quality was assessed using the validated short version of the Adolescent Sleep-Wake Scale (ASWS short version) [36,37]. This version contains 10 items measuring four dimensions of sleep quality in adolescents: going to bed; falling asleep; reinitiating sleep after waking up at night; and returning to wakefulness in the morning. Possible answers are organized in terms of frequency of experiencing sleep issues (always; almost always; often; sometimes; rarely; and never). A French version translated by a certified translator of that questionnaire was used. Our team previously used that questionnaire to measure French-speaking adolescents’ sleep quality, and the ASWS short version had adequate internal consistency (α = 0.82) [38]. Sociodemographic data (age, biological sex, gender, school level, and region) were also collected to describe the sample, and those variables were added as potential covariates in the statistical analyses.

### Statistical Analyses

The consumption of each type of SSB and water was added to obtain the mean total SSB and the mean total water intake in mL per day. The caffeine content in mg per day of each caffeinated SSB was estimated using data from the Canadian Pediatric Society [39] and adapted to the SSB measured using the BEVQ (see Table 1). A mean score of sleep quality was also calculated, with a high score indicating good sleep quality [36,37]. The internal consistency of the French version of the ASWS short version was verified by computing a Cronbach alpha coefficient (α) [40]. Using the criteria of Nunnally [41], α > 0.70 was considered adequate. No Cronbach alpha coefficient was calculated for the French version of the BEVQ, since this questionnaire measures the consumption of different beverages which might not be correlated. Descriptive statistics, including means, standard deviations, medians, interquartile ranges, and percentages, were computed.

Significant differences in beverage consumption and sleep quality between adolescent girls and boys were identified using the Wilcoxon signed-rank test given that most variables were not normally distributed. A chi-squared test was used to determine if the percentages of adolescents adhering to the public health recommendation of a maximum of 100 mg/day of caffeine [21] differed between adolescent girls and boys. A Wilcoxon signed-rank test was used to determine if adhering to the public health recommendation of a maximum of 100 mg/day of caffeine [21] was associated with better sleep quality.

Spearman correlations (r_s_) were computed to identify which beverages were significantly correlated with sleep quality in adolescents. Linear regression analyses and its percentage of variance explained (R^2^) were computed to determine if the consumption of different beverages was associated with adolescents’ sleep quality while controlling for biological sex and age. Moderation analyses were also conducted to determine if some variables that were significantly associated with adolescents’ sleep quality interacted with each other (e.g., type of beverage × biological sex) given that previous studies reported sex differences in beverage intake and sleep among adolescents. Alpha level was set at *p* < 0.05, and all analyses were performed in SAS, version 9.4 (SAS Institute, Cary, NC, USA).

## 3. Results

### 3.1. Characteristics of Participants, Beverage Consumption, and Sleep Quality

Data from three adolescents aged 13 years and two adolescents who had missing data for age were removed from the analyses. The adolescents’ ages ranged from 14 to 17 years, and the mean age was 15.3 ± 1.1 years. There was no significant difference in age between adolescent girls and boys (girls: 15.3 ± 1.1 years versus boys: 15.2 ± 1.1 years; *p* = 0.2455), and similar proportions of adolescent girls and boys were in each age category (*p* = 0.5557). A little over half of the sample (55.5%) comprised female adolescents. The characteristics of the participants are presented in Table 2. The French version of the ASWS short version had adequate internal consistency (α = 0.81) as determined using the criteria of Nunnally [41].

The adolescent boys had a significantly higher total SSB intake compared to the girls (*p* = 0.0036), including for the following types of SSBs: soft drinks (*p* = 0.0039), sports drinks (*p* = 0.0006), and flavored milk (*p* < 0.0001) (see Table 3). The adolescent boys also had a higher intake of 100% pure fruit juice (*p* = 0.0027) and carbonated flavored water (*p* = 0.0260) compared to the girls. However, the adolescent girls had a significantly higher consumption of sugar-sweetened coffee (*p* < 0.0001) compared to the boys. The main source of caffeine in the adolescent girls’ beverage intake was sugar-sweetened coffee (21.59 ± 75.37 mg/day), while in boys, it was energy drinks (4.45 ± 18.29 mg/day).

A higher proportion of adolescent boys adhered to the public health recommendation of a maximum of 100 mg/day of caffeine [21] (see Table 3), but it is worth noting that in the present sample, only nine adolescents (eight adolescent girls and one boy) did not adhere to that public health recommendation. The nine adolescents who did not adhere to the public health recommendation of a maximum of 100 mg/day of caffeine [21] appeared to have worse sleep quality (adhered to public health caffeine recommendation: 3.99 ± 0.86 versus did not adhere to this recommendation: 3.16 ± 1.00; *p* = 0.0180). The adolescent boys had a significantly better overall sleep quality compared to the girls (*p* < 0.0001), and this was also reflected in their abilities to fall asleep (*p* = 0.0002), reinitiate sleep after waking up at night (*p* < 0.0001), and return to wakefulness in the morning (*p* < 0.0001) (see Table 3).

### 3.2. Associations between Beverage Consumption and Sleep Quality

The total water (r_s_ = 0.09; *p* = 0.1630), 100% pure fruit juice (r_s_ = 0.07; *p* = 0.2910), and total SSB (r_s_ = −0.09; *p* = 0.1755) consumption levels were not significantly correlated with sleep quality in adolescents. Among caffeinated SSBs, soft drinks (r_s_ = −0.03; *p* = 0.6882), sugar-sweetened tea (r_s_ = 0.10; *p* = 0.1600), and hot chocolate (r_s_ = 0.01; *p* = 0.8944) consumption was not significantly correlated with sleep quality among adolescents, while energy drink (r_s_ = −0.16; *p* = 0.0197) and sugar-sweetened coffee (r_s_ = −0.33; *p* < 0.0001) intake was significantly and negatively correlated with adolescents’ sleep quality. Flavored milk (r_s_ = 0.14; *p* = 0.0449) was borderline significantly correlated with sleep quality in adolescents, but its association with adolescents’ sleep quality while controlling for age and sex was not verified because it could contain a mix of caffeinated flavored milk, like chocolate milk, but also strawberry or vanilla soy or almond milk that do not contain caffeine.

*Energy Drinks*. Given that energy drink intake was significantly and negatively correlated with sleep quality in adolescents, its association with adolescents’ sleep quality while controlling for age and sex was verified. Energy drink consumption was significantly and negatively associated with sleep quality in adolescents (see Model 1 for energy drinks in Table 4), including when the model was adjusted for biological sex and age (see Model 2 for energy drinks in Table 4). Biological sex was also significantly associated with adolescents’ sleep quality. This final model explained 15% of the variance in sleep quality among adolescents. The interaction of energy drink consumption × biological sex was not statistically significant (β = 0.0030; standard error (SE) = 0.0028; *p* = 0.2923).

*Sugar-Sweetened Coffee*. Sugar-sweetened coffee consumption was significantly and negatively correlated with sleep quality in adolescents. Its association with adolescents’ sleep quality while controlling for age and sex was thus verified. Sugar-sweetened coffee intake was significantly and negatively associated with sleep quality among adolescents (see Model 1 for sugar-sweetened coffee in Table 4), but this was not the case when the model was adjusted for biological sex and age (see Model 2 for sugar-sweetened coffee in Table 4). Biological sex was significantly associated with adolescents’ sleep quality. This final model explained 12% of the variance in sleep quality among adolescents.

The Interaction of sugar-sweetened coffee consumption × biological sex was statistically significant (β = −0.0211; SE = 0.0075; *p* = 0.0053). The model including this interaction explained 15% of the variance in adolescents’ sleep quality. The correlation between sugar-sweetened coffee consumption and overall sleep quality was similar among adolescent girls and boys, but this association varied based on dimensions of sleep quality between girls and boys (see Table 5). Sugar-sweetened coffee consumption was significantly and negatively correlated with adolescent girls’ abilities to go to bed (*p* = 0.0203) and fall asleep (*p* = 0.0020), while in boys it was only significantly and negatively correlated with their abilities to go to bed (*p* = 0.0069) and not their abilities to fall asleep (*p* = 0.2046). In both adolescent girls and boys, sugar-sweetened coffee consumption was not significantly correlated with their abilities to reinitiate sleep after waking up at night nor return to wakefulness in the morning.

## 4. Discussion

The results of this study suggest that there is a negative association between the consumption of specific caffeinated SSBs and sleep quality in adolescents. In the present study, no positive associations between the intake of certain beverages and adolescents’ sleep quality were found. Water intake was not significantly associated with sleep quality among adolescents, while in our previous literature review, water consumption was associated with a long sleep duration in adolescents [4]. The intake of 100% pure fruit juice was not significantly associated with sleep quality in adolescents, while in a previous study, it was associated with an earlier weekday bedtime among adolescents [5]. The total SSB consumption was also not significantly associated with sleep quality in adolescents, while previous studies reported that it was associated with a short sleep duration [6] and social jetlag [7]. These results indicate that more research is needed to identify which aspects of adolescents’ sleep (e.g., duration, quality, timing, and social jetlag) are impacted by the consumption of various beverages.

In the present study, only the intake of a few caffeinated SSBs was significantly and negatively correlated with sleep quality in adolescents. Soft drinks, sugar-sweetened tea, and hot chocolate consumption was not significantly correlated with sleep quality among adolescents, while energy drink and sugar-sweetened coffee intake was significantly and negatively correlated with adolescents’ sleep quality. A possible explanation for this is that soft drinks, sugar-sweetened tea, and hot chocolate all have lower caffeine contents compared to energy drinks and sugar-sweetened coffee according to data from the Canadian Pediatric Society [39]. The amounts consumed were also low and far below the public health recommendation of a maximum of 100 mg/day of caffeine [21]. The total caffeine content from these beverages might have been too low to significantly affect adolescents’ sleep quality. The timing of caffeine consumption is another important aspect to consider, as consuming caffeinated beverages close to bedtime is more likely to affect sleep quality [16,42,43]. Another possible explanation is related to the tool used to measure beverage consumption in the present study. The BEVQ was not designed to measure caffeine intake; therefore, the different categories of SSBs can contain a mix of caffeinated beverages and others that do not contain caffeine, such as the category for soft drinks and flavored milk. Soft drinks can include beverages with caffeine, like cola, but also lemon/lime or orange soda and ginger ale that do not contain caffeine. Flavored milks can include chocolate milk that contains caffeine, but also strawberry or vanilla soy or almond milk that do not contain caffeine. It was therefore not possible to estimate the caffeine content of caffeinated soft drinks and chocolate milk. The categories of sugar-sweetened tea and coffee might contain decaffeinated tea (e.g., herbal tea with honey) and coffee. The consumption of tea and coffee might have been underestimated, since the BEVQ only includes sugar-sweetened tea and coffee. A black coffee would therefore not be included in the category of sugar-sweetened coffee, but rather in the category of non-carbonated flavored water. However, with the increasing popularity of flavored cold (e.g., iced mochas or cappuccinos) and warm sugar-sweetened coffee (e.g., mocha coffee or vanilla flavored coffee) among adolescents in Canada [34] and the United States [28], it is unlikely that many adolescents consumed black coffee.

Energy drink consumption was significantly and negatively associated with adolescents’ sleep quality, even though the quantity consumed was lower than that for other caffeinated SSBs, such as soft drinks. A previous study in Canada reported that energy drink consumption is associated with a short sleep duration among adolescents [20]. Studies from the United States indicate a decline in caffeine consumption from soft drinks and an increase in caffeine intake from energy drinks and coffee among adolescents [44], with coffee being an important source of caffeine among adolescents [45]. In Canada, the sales per capita for non-diet soft drinks decreased by 27%, while energy drink and sugar-sweetened coffee sales per capita increased by 638% and 579%, respectively, from 2004 to 2015 [46]. In the present sample, the adolescents consumed a higher quantity of soft drinks than energy drinks and sugar-sweetened coffee, but energy drink intake is nonetheless associated with poor sleep quality among this population.

The interaction between sugar-sweetened coffee consumption and biological sex was statistically significant. Sugar-sweetened coffee intake was negatively correlated with adolescent girls’ abilities to go to bed and fall asleep, while in boys, it was only negatively correlated with their abilities to go to bed. This result suggests that sugar-sweetened coffee consumption is problematic, primarily for adolescent girls’ abilities to fall asleep. This is probably because adolescent girls in the present sample mentioned consuming a significantly higher quantity of sugar-sweetened coffee compared to boys. If adolescent boys had consumed higher amounts of sugar-sweetened coffee, their intake most likely would have also been negatively associated with their abilities to fall asleep. The present results also indicate that adolescent girls are more at risk for poor sleep quality than boys, even though both were of similar ages. Adolescent girls had lower scores rating their abilities to fall asleep, to reinitiate sleep after waking up at night, and to return to wakefulness in the morning. This is in line with previous studies reporting that adolescent girls usually have poorer sleep quality compared to boys [16,38]. In a previous study among adolescents in New Zealand, adolescent girls had poorer sleep quality and consumed more warm caffeinated beverages, such as coffee, tea and hot chocolate, after dinner than boys [16]. In the present study, sex differences in sleep quality cannot be explained by age and maturation differences between adolescent girls and boys like in previous studies [22,29]. This suggests that behavioral reasons are at play, such as adolescent girls consuming more caffeinated SSBs, mainly from sugar-sweetened coffee, than boys.

There were important sex differences in beverage and caffeine intake. Similarly to previous studies [23,24], adolescent boys had a significantly higher total SSB intake compared to girls. The main source of caffeine in adolescent girls’ beverage intake was sugar-sweetened coffee, while in boys, it was energy drinks. Previous studies reported that adolescent boys have a higher consumption of energy drinks compared to girls [16,19,25], and the results from a study conducted in New Zealand indicated that adolescent girls consume more warm caffeinated beverages, such as coffee, tea and hot chocolate than boys [16]. Coffee and energy drinks are perceived to be the most popular caffeinated beverages by adolescents themselves, and they are associated with social status and fitting in [47,48]. The present results suggest that consuming energy drinks might be more widespread among adolescent boys, while drinking sugar-sweetened coffee might be more common in girls. A higher proportion of adolescent boys adhered to the public health recommendation of a maximum of 100 mg/day of caffeine [21]. However, only nine adolescents (eight adolescent girls and one boy) did not adhere to that public health recommendation in the present sample. Adhering to the public health recommendation of a maximum of 100 mg/day of caffeine [21] appeared to be associated with better sleep quality in adolescents, but these results need to be replicated among a larger sample of adolescents who consume caffeine.

### Strengths and Limitations

The present study has a few strengths and limitations. The main strength is using French versions [34,38] of validated questionnaires measuring the intake of various beverages (SSBs, 100% pure fruit juice, and water) [35] and sleep quality [36,37] specifically designed for adolescents. A novel aspect of this study is the focus on adolescents’ sleep quality, while most of the research on the association between diet and sleep among adolescents seems to be focused on sleep duration [30]. The caffeine contents of caffeinated SSBs were estimated using data from the Canadian Pediatric Society [39]. However, it was not possible to estimate the caffeine content of caffeinated soft drinks and chocolate milk. The estimated total daily caffeine intake is therefore most likely underestimated, and this could explain why a high proportion of adolescents adhered to the public health recommendation of a maximum of 100 mg/day of caffeine [21]. The main limitation of this study is that the BEVQ was not designed to measure caffeinated SSB consumption. It would be better to confirm the present results with a validated questionnaire specifically developed to measure caffeine intake, such as the Caffeine Consumption Habits Questionnaire [49], although this questionnaire was originally created to measure adults’ caffeine intake, and not that of adolescents’. Another limitation is that this is a cross-sectional study, so a causation between caffeinated SSBs and sleep quality cannot be established, and there is the possibility of a bidirectional effect of caffeine on sleep in children and adolescents [50]. It is possible that adolescents who have poor sleep quality compensate for their lack of energy the next day by consuming more caffeinated SSBs. The sample comprised volunteers and might not be representative of adolescents in the province of Québec (Canada). Future studies should ideally use a longitudinal design, measure the timing of adolescents’ caffeine consumption in relation to their bedtimes, evaluate the impact of other potential confounders (e.g., screen use [38] and physical activity [33]), and use a combination of self-reported and objective measures of adolescents’ sleep quality to overcome the limitations associated with both types of measures [51] (e.g., memory bias for self-reported and reactivity for objective measures).

## 5. Conclusions

Consuming specific caffeinated SSBs, such as energy drinks and sugar-sweetened coffee, is negatively associated with sleep quality in adolescents. Energy drink consumption is negatively associated with adolescents’ sleep quality, while sugar-sweetened coffee intake is negatively associated with adolescent girls’ abilities to go to bed and to fall asleep, and it is only associated boys’ abilities to go to bed. Adolescent girls are more at risk for poor sleep quality compared to boys. There are important sex differences in beverage intake and sleep quality, suggesting the need to adapt public health interventions promoting sleep quality depending on whether they are aimed at adolescent girls or boys. Public health interventions should suggest adolescents, especially girls who are more at risk for poor sleep quality, to limit their consumption of caffeinated SSBs. Public health interventions aimed at adolescent boys should primarily target lowering energy drink consumption, while those aimed at girls should target sugar-sweetened coffee intake to possibly improve their sleep quality as well as promote their physical and mental health.

## Figures and Tables

**Table 1 nutrients-16-00285-t001:** Estimated caffeine content of caffeinated sugar-sweetened beverages.

Beverage	Serving Size		Approximate Caffeine Content (mg) *	Approximate Caffeine Content Used in the BEVQ (mg/Serving and mg/mL)
	oz	mL		
Energy drinks	8	237	95	Energy drinks: 95 mg/serving and 0.40 mg/mL
Instant coffee	8	237	76–106 (91)	Sugar-sweetened coffee: 119.75 mg/serving and 0.51 mg/mL
Roasted and ground brewed coffee	8	237	118–179 (148.5)	
Black tea	8	237	43	Sugar-sweetened tea: 36.5 mg/serving and 0.15 mg/mL
Green tea	8	237	30	
Regular cola beverage	12	355	36–46	N/A Regular or non-diet soft drink intake is measured, and not just cola
Chocolate milk	8	237	8	N/A Flavored milk intake is measured, and not just chocolate milk
Hot chocolate	8	237	5	Hot chocolate: 5 mg/serving and 0.02 mg/mL

Note. BEVQ: Beverage Questionnaire; N/A: data not available in the BEVQ. * Source: Canadian Pediatric Society [39].

**Table 2 nutrients-16-00285-t002:** Characteristics of participants (n = 218).

Variables	n	%
*Age*		
14 years	74	33.9
15 years	54	24.8
16 years	51	23.4
17 years	39	17.9
*Biological sex*		
Female	121	55.5
Male	97	44.5
*Gender*		
Girl	121	55.5
Boy	95	43.6
Neither a girl nor a boy	0	0
I prefer not to answer	2	0.9
Other	0	0
*School level*		
2nd year of high school (13–14 years)	52	23.9
3rd year of high school (14–15 years)	49	22.5
4th year of high school (15–16 years)	59	27.1
5th year of high school (16–17 years)	43	19.7
Other	15	6.9

**Table 3 nutrients-16-00285-t003:** Beverage consumption and sleep quality in adolescents (n = 218).

Variables	Girls (n = 121)			Boys (n = 97)			*p*-Value *
	Mean ± SD or %	Median (IQR)	Estimated Caffeine Content (mg/day)	Mean ± SD or %	Median (IQR)	Estimated Caffeine Content (mg/day)	
*Total SSBs (mL/day)*	**256.84 ± 324.60**	165.00 (71.43–286.43)	25.84 ± 77.54	**316.27 ± 274.14**	267.86 (107.14–400.00)	6.91 ± 21.69	**0.0036**
Soft drinks	**42.05 ± 114.81**	0 (0–35.71)	N/A	**51.30 ± 80.06**	35.71 (0–50.71)	N/A	**0.0039**
Fruit drinks	76.28 ± 125.51	28.57 (0–89.29)	0	69.56 ± 120.11	35.71 (0–89.29)	0	0.9393
Energy drinks	7.23 ± 35.23	0 (0–0)	2.89 ± 14.09	11.13 ± 45.72	0 (0–0)	4.45 ± 18.29	0.3316
Sports drinks	**30.51 ± 66.67**	0 (0–17.14)	0	**64.60 ± 96.67**	0 (0–89.29)	0	**0.0006**
Sugar-sweetened tea	8.21 ± 30.81	0 (0-0)	1.23 ± 4.62	8.54 ± 31.80	0 (0-0)	1.28 ± 4.77	0.9215
Sugar-sweetened coffee	**42.33 ± 147.78**	0 (0–35.71)	21.59 ± 75.37	**1.99 ± 11.05**	0 (0–0)	1.01 ± 5.64	**<0.0001**
Vitamin water	17.50 ± 138.12	0 (0–0)	0	24.99 ± 130.55	0 (0–0)	0	0.2185
Slushies	8.44 ± 20.31	0 (0–0)	0	6.88 ± 16.89	0 (0–0)	0	0.5741
Flavored milk	**17.56 ± 35.36**	0 (0–28.57)	N/A	**69.07 ± 118.42**	28.57 (0–89.29)	N/A	**<0.0001**
Hot chocolate	6.74 ± 18.07	0 (0–0)	0.13 ± 0.36	8.22 ± 25.89	0 (0–0)	0.16 ± 0.52	0.9171
*100% pure fruit juice (mL/day)*	**58.39 ± 76.77**	28.57 (0–85.72)	0	**131.97 ± 234.98**	71.43 (0–178.57)	0	**0.0027**
*Total water (mL/day)*	1656.68 ± 1620.89	1065.00 (710.00–2100.01)	0	1352.26 ± 1269.85	1000.00 (539.29–1800.00)	0	0.3457
Non-carbonated non-flavored water	1443.10 ± 1408.09	1000.00 (600.00–1800.00)	0	1135.93 ± 865.13	750.00 (500.00–1500.00)	0	0.3435
Non-carbonated flavored water	200.77 ± 569.40	0 (0–71.43)	0	239.28 ± 652.83	0 (0–89.29)	0	0.8375
Carbonated non-flavored water	40.69 ± 184.23	0 (0–0)	0	34.92 ± 120.54	0 (0–35.71)	0	0.0730
Carbonated flavored water	**27.90 ± 131.79**	0 (0–0)	0	**30.37 ± 88.11**	0 (0-0)	0	**0.0260**
*Adhere to the recommendation of a maximum of 100 mg/day of caffeine, %*	**93.4**			**99.0**			**0.0396**
*Total sleep quality score* ^†^	**3.70 ± 0.86**			**4.27 ± 0.79**			**<0.0001**
Going to bed (3 items)	3.50 ± 1.14			3.74 ± 1.11			0.1605
Falling asleep (2 items)	**3.69 ± 1.43**			**4.40 ± 1.44**			**0.0002**
Reinitiating sleep (3 items)	**4.44 ± 1.24**			**5.13 ± 0.97**			**<0.0001**
Returning to wakefulness (2 items)	**2.91 ± 1.20**			**3.66 ± 1.17**			**<0.0001**

Note. SD: standard deviation; IQR: interquartile range; SSBs: sugar-sweetened beverages; N/A: data not available. ^†^ Possible range of scores is 1–6, with higher scores indicating good sleep quality. * Numbers in bold are statistically significant (*p* < 0.05).

**Table 4 nutrients-16-00285-t004:** Association between caffeinated sugar-sweetened beverage consumption and sleep quality (n = 218).

Variables	β Coefficient (SE) and *p*-Value *	
	*Model 1*	*Model 2*
*Energy drinks*		
Energy drink consumption (mL/day)	**−0.0046 (0.0015)** **0.0020**	**−0.0048 (0.0014)** **0.0005**
Biological sex (male )		**0.6033 (0.1105)** **<0.0001**
Age (years)		0.0724 (0.0496) 0.1457
Adjusted R^2^	0.04	0.15
Model F (*p*-value)	**9.79 (0.0020)**	**13.99 (<0.0001)**
*Sugar-sweetened coffee*		
Sugar-sweetened coffee consumption (mL/day)	**−0.0014 (0.0005)** **0.0095**	−0.0010 (0.0005) 0.0507
Biological sex (male)		**0.5463 (0.1143)** **<0.0001**
Age (years)		0.0846 (0.0506) 0.0961
Adjusted R^2^	0.03	0.12
Model F (*p*-value)	**6.86 (0.0095)**	**10.75 (<0.0001)**

Note. β: standardized beta coefficient; SE: standard error;. * Numbers in bold are statistically significant (*p* < 0.05).

**Table 5 nutrients-16-00285-t005:** Correlations between sugar-sweetened coffee consumption and sleep quality in adolescents by biological sex (n = 218).

Variables	Spearman Correlations (*p*-Value) *	
	Girls (n = 121)	Boys (n = 97)
*Overall sleep quality*	**−0.25 (0.0053)**	**−0.24 (0.0200)**
Going to bed	**−0.21 (0.0203)**	**−0.27 (0.0069)**
Falling asleep	**−0.28 (0.0020)**	−0.13 (0.2046)
Reinitiating sleep	−0.07 (0.4616)	−0.09 (0.3922)
Returning to wakefulness	−0.16 (0.0869)	−0.09 (0.3687)

Note. * Number in bold are statistically significant (*p* < 0.05).

## Data Availability

The data presented in this study are available on request from the corresponding author. The data are not publicly available due to the Research Ethics Committee of the CISSS de Chaudière-Appalaches did not approve the public sharing of our dataset.

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
