# Peer review of "Association between Beverage Consumption and Sleep Quality in Adolescents"

_nutrients, 2024, doi:10.3390/nu16020285_

Round 1

Reviewer 1 Report

Comments and Suggestions for Authors

Overall relevance:

A relevant research paper, addressing an important topic that affects our younger generation and is also very relevant, as the market for these drinks has exploded in recent years and young people, and in some cases even children, are being specifically targeted.

However, the points below led to a weakening of the paper, especially the Methods section. These should be carefully considered and improvements made where possible.

Major comments:

Material and Methods:

L-93/94: Inclusion criterion of age 14 to 18 years defined. Why did you choose this range and not the ages of e.g. 12 to 18?. In the results part (L-166) you showed an age range from 13 to 17 years, which is not consistent with the inclusion criteria.

L-128/130: Inclusion of confounding variables. The list of confounding variables is very minimal. I would suggest considering the following potential factors (if available): BMI, physical activity and media consumption, as these are important factors that may also impact sleep behavior and sleep quality.

If these data are not available, these factors should be mentioned in the discussion as potential confounders.

L-112/119: Why didn’t you assess light (artificially sweetened) beverages? Since the proportion of light drinks has increased substantially in the past years it would be interesting to compare between sugar-sweetened beverages and light beverages.

Minor comments:

Discussion:

L-325/345 (Strengths and Limitations): discuss whether there might be a selection bias in your study sample or how representative this sample is.

Author Response

Overall relevance

We thank the reviewer for this kind comment and his/her suggestions to improve our paper.

Major Comments

Material and Methods

1) We decided to include adolescents aged 14 to 18 years as in the province of Québec in Canada (i.e., where the study was conducted), 14 years is the age at which a minor can accept to participate in a study of minimal risk (like our study) without asking his/her parents for permission. When we looked at the data, we noticed that 3 adolescents who were 13 years old decided to answer our survey. We decided to keep their data in our analyses given that they are adolescents and had complete data.

2) We agree that those are important confounding variables for sleep. However, we do not have this data in the present study. Our goal was to keep the survey as short as possible to favor participation from adolescents. We added in the discussion that future studies should ideally evaluate the impact of other potential confounders such as screen use and physical activity and added references to support this claim (see lines 351-352 on page 10).

3) We agree that diet/light (artificially sweetened) beverages have gained in popularity among adolescents in recent years. There is currently a controversy in Canada concerning whether they are considered sugar-sweetened beverages (SSB) since they do not contain any added (real) sugar and calories, and therefore are less likely to contribute to overweight/obesity in adolescents. Our definition of SSB is from The CDC Guide to Strategies for Reducing the Consumption of Sugar-Sweetened Beverages (2010) (https://stacks.cdc.gov/view/cdc/51532) and diet/light (artificially sweetened) beverages are not considered SSB in that document.

Minor Comments

Discussion

4) We added that another limitation of our study is that our sample was comprised of volunteers and therefore it might not be representative of adolescents in the province of Québec (Canada) (see lines 348-349 on page 10).

Reviewer 2 Report

Comments and Suggestions for Authors

This paper examines the relationship between consumption of various beverages and sleep quality among adolescents.  Although the paper is generally well-written, the very small effect sizes call into question the meaningfulness of findings.  Additionally, the Results section is missing key details.  Specific concerns are listed below.

1.     I agree that it is a strength to assess sleep quality, given the lack of existing research, but it would be helpful to also report associations with sleep duration.  Was sleep duration data collected?  I suggest reporting these findings if so (even if the results are null).

2.     A table displaying correlations between all study variables would provide a more complete and clear summary of key associations.

3.     The reported standardized beta coefficients linking beverage consumption to sleep quality are extremely small (all less than 0.01; standardized betas range from 0-1, with 0 indicating no association).  Even if these effects are statistically significant, it is questionable whether the effects are meaningful.

4.     More details are needed in the Results section regarding the significant interaction between coffee consumption and sex.  Post-hoc probing should be used to determine whether coffee consumption was associated with sleep quality for either boys and/or girls.  Please provide beta coefficients for girls and boys separately (or a figure depicting the interaction, etc., as long as results are made clear).  The validity of the interpretation of the interaction in the Discussion section is unclear, given the absence of reported findings.

5.     The opening of the Discussion states how findings contradict previous research.  Any potential explanations for the discrepant findings?

6.     Subjective measurement of sleep is an additional limitation of the study.  Replication with objective measures (e.g., actigraphy) would be useful for a better understanding of associations between beverage consumption and sleep quality.

Author Response

We thank the reviewer for this kind comment and his/her suggestions to improve our paper.

1) We agree that it would be nice to also report sleep duration. However, we did not collect that data. The short version of the Adolescent Sleep-Wake Scale does not measure sleep duration. Our goal was solely to measure sleep quality to differentiate ourselves from previous studies that focused on adolescents’ sleep duration.

2) All the correlations between each beverage and sleep quality are presented in the paragraph entitled “Associations between Beverage Consumption and Sleep Quality” on lines 202-209 on page 7. We already have four tables in the article, so we believe that this information does not need to be reported in a table.

3) We agree that the standardized beta coefficients linking beverage consumption to sleep quality are extremely small. It is however important to bear in mind that we ran linear regression analyses and that our unit for beverages is in ml/day, indicating that the standardized beta coefficients report the association between an increase of 1 ml/day of energy drinks or sugar-sweetened coffee and sleep quality. If we had used logistic regression analyses and dichotomized our beverage intake at a higher amount (e.g., 1 cup or 250 ml/day) the association most likely would have been stronger as consuming 250 ml instead of 1 ml is more likely to affect adolescents’ sleep quality. We decided to treat beverage intake continuously to avoid losing statistical power when dichotomizing a continuous variable. All our linear regression models are statistically significant and explain between 3 to 16 percent of the variance (see Table 4 on page 8).

4) We added in the Results section the beta coefficients and standard error (SE) for the two interactions we tested (energy drinks × biological sex; and sugar-sweetened coffee × biological sex) (see lines 218 and lines 232-233 on pages 7-8). We added a figure to depict the interaction (see Figure 1) and a sentence stating that adolescent girls had a higher consumption of sugar-sweetened coffee and reported poorer sleep quality compared to boys to explain the meaning behind this result (see lines 233-234 on page 8). As already stated in the discussion on lines 294-299 on page 9, the interaction between sugar-sweetened coffee and biological sex suggests that sugar-sweetened coffee intake is problematic primarily for adolescent girls’ sleep quality. This is probably because adolescent girls in the present sample mentioned consuming a significantly higher quantity of sugar-sweetened coffee compared to boys. If adolescent boys had consumed higher amounts of sugar-sweetened coffee, their intake most likely would have also been negatively associated with their sleep quality. We were careful in our interpretation of this result as we did not want this to be interpreted that sugar-sweetened coffee consumption cannot have a negative impact on adolescent boys’ sleep quality.

5) We believe that we did not find an association between water, 100% pure fruit juice, total SSB intake, and sleep quality in our study because previous studies have measured other components of adolescents’ sleep (duration, timing, and social jetlag). This is why we are suggesting at the end of that opening paragraph of our discussion that more research is needed to identify which aspects of adolescents’ sleep are impacted by the consumption of various beverages (see lines 247-250 on page 9).

6) We agree with the reviewer and added in the discussion that future studies should ideally use a combination of self-reported and objective measures of adolescents’ sleep quality to overcome limitations associated with both types of measures (e.g., memory bias for self-reported and reactivity for objective measures) and added a reference to support this statement (see lines 352-354 on pages 10-11).

Round 2

Reviewer 2 Report

Comments and Suggestions for Authors

With this revision, the authors have adequately addressed much of the feedback on the original manuscript.  However, I still have major concerns with the interpretation of results, described in detail below.

Regarding previous comment #3 concerning the small regression coefficients, it appears that the authors are discussing unstandardized coefficients, which represent associations in the mL unit, rather than standardized beta coefficients, which are interpreted similarly to a correlation coefficient.  If the coefficients reported in the manuscript are indeed unstandardized, the description of “standardized” should be removed from the Table 4 note.  Additionally, “B” is typically used to represent unstandardized coefficients, whereas β typically represents standardized betas.

Previous comment #4 was also not sufficiently addressed.  Although I appreciate the addition of the regression coefficients for the interaction terms, this does not allow for sufficient interpretation of the interaction effect.  The authors have provided an interpretation of the two main effects:  that is, that girls reported greater coffee consumption than boys and that girls reported lower sleep quality than boys.  Figure 1 also seems to depict the two main effects, rather than the interaction.  An interaction effect implies that the association between coffee consumption and sleep quality is different for boys and for girls.  However, it is unclear if the association between coffee consumption and sleep quality is stronger for girls or boys, and it is unclear whether this association is statistically significant for girls, boys, or both.  Without this information, it is impossible to evaluate the validity of the interpretation stated in the Discussion section.  The authors should conduct post-hoc probing of the interaction effect to determine the slope of the association between coffee consumption and sleep quality separately for boys and girls.  I recommend deleting Figure 1 or replacing it with a figure depicting the separate slopes of the association for girls and boys.

Author Response

1. We are only reporting standardized beta coefficients in the article which is why we are using the Greek lowercase letter for beta (β).

2. We removed Figure 1 and added a new table (see Table 5) presenting correlations between sugar-sweetened coffee consumption and adolescents’ sleep quality by biological sex. This new analysis shows that the correlation between sugar-sweetened coffee consumption and overall sleep quality is similar among adolescent girls and boys, but this association varies by dimensions of sleep quality between girls and boys. Sugar-sweetened coffee consumption was significantly and negatively correlated with adolescent girls’ ability to go to bed and fall asleep while in boys it was only significantly and negatively correlated with their ability to go to bed. This is now mentioned in the results section (see lines 243-251 on page 10). We modified the abstract while respecting the maximum of 200 words (see abstract on page 1), the discussion (see lines 311-320 on page 12), and the conclusion (see lines 380-381 on page 13) accordingly.
